

# Seasonal and inter-annual community structure characteristics of zooplankton driven by water environment factors in a sub-lake of Lake Poyang, China

Beijuan Hu[1], Xuren Hu[1], Xue Nie[1], Xiaoke Zhang[2], Naicheng Wu[3], Yijiang Hong[1] and Hai Ming Qin[4]

[1] School of Life Science and Center for Watershed Ecology of Institute of Life Science, Nanchang University, Nanchang, China
[2] Research Center of Aquatic Organism Conservation and Water Ecosystem Restoration in University of Anhui Province, Anqing Normal University, Anqing, China
[3] Aarhus Institute of Advanced Studies, Aarhus University, Aarhus, Denmark
[4] School of Life Science and Center for Watershed Ecology of Institute of Life Science; School of Life Sciences; Jiangxi Province Key Laboratory of Watershed Ecosystem Change and Biodiversity, Nanchang University; Qufu Normal University, Qufu, China

Corresponding authors
Yijiang Hong, yjhong2008@163.com
Hai Ming Qin,
qinhaiming-haha@163.com,
qinhaiming@ncu.edu.cn

## ABSTRACT

**Background**. Sub-lakes are important for the maintenance of the ecosystem integrity of Lake Poyang, and zooplankton play an important role in its substance and energy flow.

**Methods**. A seasonal investigation of zooplankton was conducted in spring (April), summer (July), autumn (October) and winter (January of the following year) from 2012 to 2016 in a sub-lake of Lake Poyang. The aim of the present study was to understand the seasonal dynamics and interannual variation of zooplankton communities and their relationship to environmental factors.

**Results**. A total of 115 species were identified in all samples in the four years, which comprised of 87 Rotifera, 13 Cladocera and 15 Copepoda. Rotifera was the dominant group in terms of quantity, and its species richness and abundance were significantly higher when compared to Cladocera and Copepoda ($P < 0.05$), while Cladocera dominated in terms of biomass. The species richness of Rotifera exhibited a significant seasonal difference ($P < 0.05$). Both the density and biomass of zooplankton revealed significant seasonal differences ($P < 0.05$). In general, the density and biomass of zooplankton were higher in summer and autumn, when compared to winter and spring. Biodiversity indices were dramatically lower in spring than in the other seasons. The non-metric multidimensional scaling (NMDS) analysis suggested that these zooplankton communities can be divided into three groups: spring community, summer-autumn community, and winter community. The seasonal succession of zooplankton communities did not have interannual reproducibility. In high water level years, the dominant species of zooplankton (Cladocerans and Copepods) in the wet season had a lower density, and the result in low water level years was exactly the opposite. The redundancy analysis revealed that water temperature (WT), conductivity, pH and dissolved oxygen (DO) had significant effects on the zooplankton community.

**Conclusions**. The community structure of zooplankton has a significant seasonal pattern, but has no interannual repeatability. In high water level years, the dominant

species of zooplankton (Cladocerans and Copepods) in the wet season had a lower density, and the result in low water level years was exactly the opposite. The density, biomass and diversity indices of zooplankton were significantly different in different seasons. The present study was helpful in the further understanding of the ecosystem stability of lakes connected with rivers, providing scientific guidance for the protection of lake wetlands.

# INTRODUCTION

Lake Poyang is the largest freshwater lake in China. It is a connected lake, in which water levels fluctuate widely during different seasons (*Wu, 1994*). In Lake Poyang's low water period, more than 100 separated sub-lakes appear (*Hu et al., 2015*). When these sub-lakes connect with the main lake in the high water period, a close exchange of material, energy and biology occur among these water bodies. Sub-lakes are of significant ecological value due to their huge vegetation biomass (*Huang & Guo, 2007*; *Li & Liu, 2001*), high biodiversity (*Wu, 1994*; *Ge et al., 2010*), fish nurseries and reproduction sites in the high water period (*Zhang & Wang, 1982*), and ideal habitats provided for wintering birds (*Qi et al., 2011*; *Hu, Ge & Liu, 2014*). All these characteristics play important and unique roles in maintaining the biological integrity and species diversity of the Lake Poyang wetland ecosystem.

Zooplanktons are essential for maintaining the health and stability of aquatic ecosystems, acting as a link between the primary producer and higher consumers. The trophic state of lakes can also be accurately reflected by the spontaneous variation in zooplankton (*Pereira et al., 2002*; *Krylov, 2015*). Zooplankton communities have significant seasonal fluctuations under the influence of biotic and abiotic factors. Environmental factors such as total nitrogen (TN), total phosphorus (TP), water temperature (WT), water clarity and the biomass of microalgae all play important roles in the succession of zooplankton communities (*Yang et al., 2014*; *Hu, Yang & Liu, 2014*). The periodical connection between lakes and rivers also affects the ecological structure and function of zooplankton communities. Different water levels have different degrees of effect on zooplankton (*Goździejewska et al., 2016*). Planktivorous fish exert high top-down control on zooplankton, especially on macro-zooplankton, which may lead to a decrease in the number of Daphnia (*Scheffer et al., 1997*) and the miniaturization of the zooplankton community.

The increase of N and P levels in recent decades (*Lv, 1996*; *Wang, Zhou & Hu, 2008*) has led to the eutrophication of Lake Poyang. In 2011, the *Jiangxi Water Resources Bulletin* (2012–2015) indicated that the water of Lake Poyang exhibited moderate eutrophication (http://www.jxsl.gov.cn/) when the average TN was 1.389 mg/L and TP was 0.067 mg/L (*Chen et al., 2013*). Although Lake Poyang has reached the level of eutrophication, there was fortunately no outbreak of cyanobacteria bloom due to both the connection of the

lake with the Yangtze River and seasonal fluctuations in water level (*Hu & Zhu, 2014*). However, cyanobacterial blooms have already taken place in Lake Poyang's sub-lakes due to unmanaged development and resource utilization (*Dai et al., 2015*). The state of the sub-lakes reflect the environmental deterioration of Lake Poyang, and their ecological decline may eventually seriously affect the lake's wetland ecosystem and function.

The first study of zooplankton in Lake Poyang focused on the species in the 1960s (*Deng, Li & Cheng, 1963*). Subsequently, other discontinuous research on zooplankton have been conducted, but these studies were relatively limited. For example, *Xie, Li & Li (1997)*, *Xie & Li (1998)* and *Xie, Li & Peng (2000)* carried out an annual dynamic research on zooplankton in Lake Poyang; *Wang et al. (2003)* catalogued 150 zooplankton species in the spring and winter; and *Liu et al. (2016)* characterized the features of all crustaceans. According to records (*Huang & Guo, 2007*), there is a total of 207 zooplankton species in China but reports on zooplankton in the sub-lakes of Lake Poyang have remained very limited in the past decades. Zooplankton in sub-lakes were reported only in recent years (*Zhang et al., 2014*). Furthermore, there are very few reports on seasonal and annual variations in zooplankton communities in sub-lakes. Zooplankton are the main feeding target of many fishes, and their distribution and variation can be used as a scientific basis for exploring fish stocks and finding fishing grounds (*Huang et al., 2010*). At the same time, zooplankton are also important indicators of aquatic environmental change (*Peter, Sigrid & Shuhei, 2010*). Their population structure, quantity, and dominant species can be important indicators for monitoring water quality (*Wang et al., 2012*). Therefore, studying the spatial and temporal distribution pattern of zooplankton can provide a vital scientific basis for the protection and sustainable utilization of lake resources.

The present study carried out a preliminary research of the seasonal variations of zooplankton communities in Shahu Lake, a sub-lake of Lake Poyang. The samples were collected seasonally from April 2012 to January 2016. The specific aims were as follows: (1) to investigate the seasonal and interannual variations of zooplankton communities in a sub-lake, and (2) to identify the dominant physicochemical factors that affect the variation in zooplankton community structures.

## MATERIALS & METHODS

### Sampling site

Lake Poyang (28°24′–29°46′N, 115°49′–116°46′) is located downstream of the Yangtze River (Fig. 1A). It has an area of 3,210 km$^2$ in the highest water level period and 146 km$^2$ in the lowest water level period (*Zhang, 1988*). Its catchment has a subtropical monsoon climate with an average annual rainfall of 1,400–1,900 mm, and an average annual temperature of 16.7–17.7 °C (*Wu, 1994*). Jiangxi Poyang Lake National Nature Reserve lies to the northwest of Lake Poyang (Fig. 1B), and there are nine sub-lakes in the reserve. One of these is Shahu Lake, which has a surface area of 1.4 km$^2$, a flat bottom, and few submerged plants (Fig. 1C). There is significant seasonal water level fluctuation in Shahu Lake. The maximum water fluctuation amplitude is approximately 6 m between flood and dry seasons. During the dry season, local fishermen fish in the lake using a
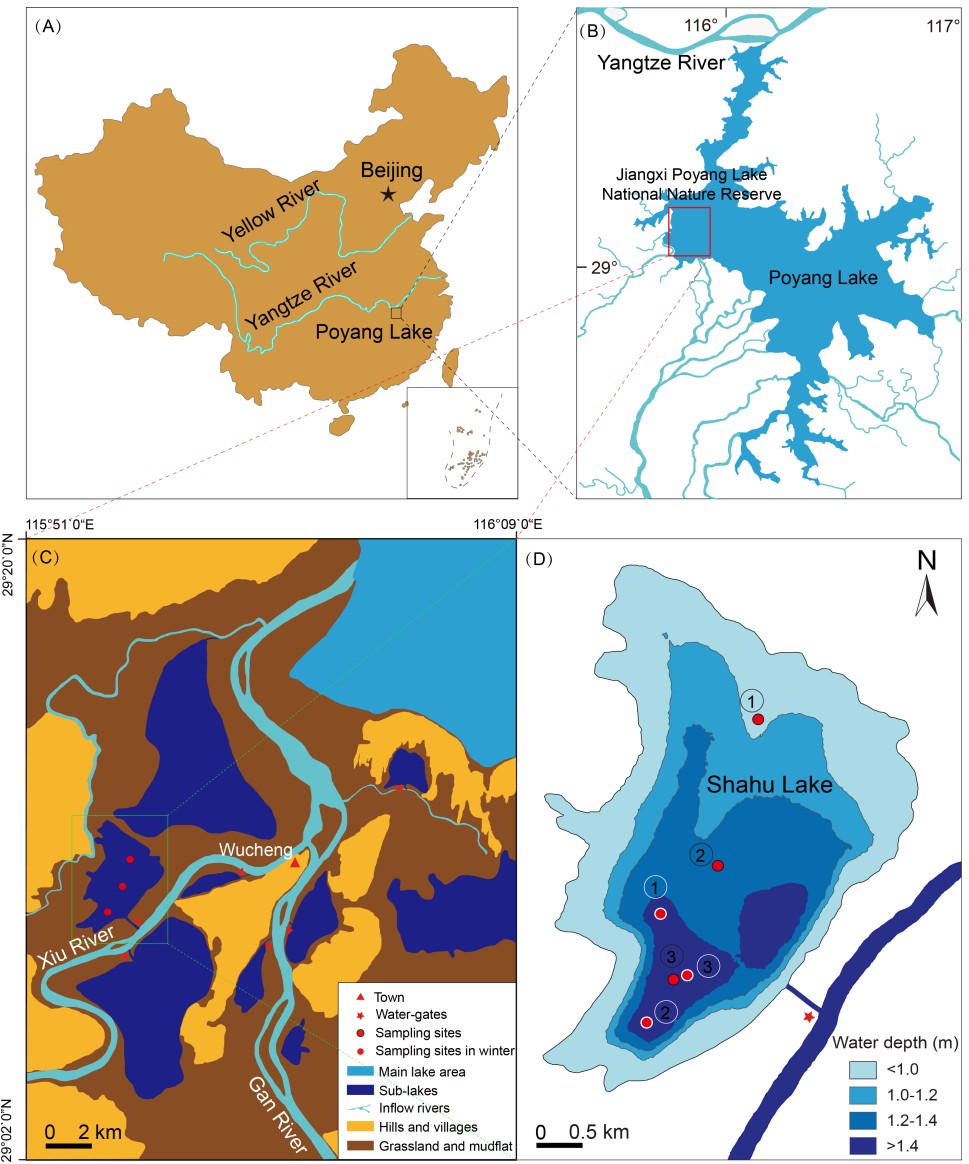

**Figure 1** **Location of Shahu Lake and the zooplankton sampling points (water depth map based on the water level of October 2012).** (A) Location of the Poyang Lake. (B) Location of Jiangxi Poyang Lake National Nature Reserve. (C) Location of Shahu Lake. (D) Zooplankton sampling points in Shahu Lake.

method known as "lake enclosed in autumn". This involves the fishermen discharging water through a water-gate, and fishing with a long mesh bag fixed at its gate from October to January of the next year. Using this process, the water level gradually decreases to 0.2–0.3 m.

## Sampling design

Zooplankton were seasonally sampled (spring = April, summer = July, autumn = October and winter = January) at three points in Shahu Lake from April 2012 to January 2016. With the water level declining, the water only remained in the deepest area, and the water depth was approximately only 0.2–0.3 m. Therefore, these three sampling points in winter were set in areas of the lake where the water depth was more than 1.4 m in autumn 2012 (Fig. 1D). Zooplankton were sampled three times at each point, and nine samples were collected in each season, resulting in a total of 144 samples over four years. A 5-L modified Schindler–Patalas sampler was used to collect 10 L of mixed water at approximately 50 cm below the water surface for each sample. A plankton net (mesh size, 64 µm) was used to filter the water and collect the zooplankton, which gathered at the end of the net, and these were immediately preserved in 50 mL plastic bottles with 4% formalin. In the laboratory, the zooplankton were counted and identified under a microscope (Olympus SZ61 and Olympus CX23; Olympus, Tokyo, Japan). When there were excessive individuals in one sample, a sub-sample method was used to estimate the actual quantity. In the present study, copepod nauplii were considered as one taxon. Four bibliographies, including three faunas, were used for zooplankton identification (*Wang, 1961*; *Crustacean Research Group, 1979*; *Jiang & Du, 1979*; *Han & Shu, 1995*).

At the time of collection, the physicochemical parameters of WT, pH, conductivity (Cond), dissolved oxygen (DO) and turbidity (Turb) were simultaneously measured using a multi-function water quality monitor (YSI6600V2; Xylem, Rye Brook, NY, USA).

## Data analysis

In the present study, zooplankton community characteristics mainly include the dominant species, diversity index, density, biomass and community clustering map.

The dominance index was calculated as follows:

$$Y = n_i \times f_i / N. \tag{1}$$

In which $Y$ represented the dominance index, $n_i$ represented the individual number of $i$ species, $f_i$ represented the occurrence frequency of $i$ species, and $N$ represented the total number of individuals. When $Y$ was greater than or equal to 0.02, this species was defined as dominant species. In the present study, $N$ referred to the total density of zooplankton in each season (*Wen et al., 2015*).

The Shannon–Weiner diversity index ($H'$), Margalef richness index ($D$) and Pielou evenness index ($J'$) calculation formulae were as follows:

$$H' = -\sum P_i \ln(P_i)$$
$$D = (S-1)/\ln N$$
$$J' = H'/\ln S() \tag{2}$$

where $S$ represented the species number and $P_i$ represented the proportion of $i$ species densities in the total zooplankton density in the sample (*Wen et al., 2015*).

The densities of zooplankton were calculated by dividing the individual numbers of zooplankton that gathered in each collection by the sample volume, and this was expressed in ind./L. The biomass of zooplankton (wet weight) was evaluated according to the method reported by *Zhang & Huang (1991)*. The weight of each nauplii was estimated to be approximately 0.003 mg (*Xie & Li, 1998*).

The seasonal variance of water physicochemical factors, zooplankton density and biomass were analysed by one-way ANOVA, using STATISTICA 7.0 (StatSoft Inc., Tulsa OK, USA). The seasonal variation in zooplankton communities was tested by non-metric multidimensional scaling analysis (NMDS) and analysis of similarities (ANOSIM). Zooplankton individual number data were analysed using a ranked similarity matrix based on Bray–Curtis similarity measures. Rare species, which had an average density of less than 1.0 ind./L, were excluded during NMDS and ANOSIM analyses. NMDS ordination and ANOSIM analyses were performed with the PRIMER 5 computer package (*Clarke & Warwick, 1994*). The indicator value method (IndVal) was used to detect how strongly each species discriminated among the NMDS groups. The indicator value of a taxon varied from 0 to 100, and the indicator value attained its maximum value when all individuals of a taxon occurred at all sites within a single group (*Szulc, Szulc & Kruk, 2010*). The significance of the indicator value for each species was tested with a Monte Carlo randomization procedure with 1,000 permutations. IndVal was performed using the *indval* function in R package *labdsv* (R version 3.4.1; *R Development Core Team, 2017*).

The correlation between water physicochemical factors and zooplankton dominant species was analysed using redundancy analysis (RDA), and the significance was determined using the Monte Carlo test. The RDA and Monte Carlo tests were performed using Canoco for Windows 4.5 software (*Ter Braak & Smilauer, 2002*). With the exception of the NMDS analysis, all variables were transformed by $\ln(x + 1)$ prior to analysis.

## RESULTS

### Physical-chemical variables

The mean seasonal values of physicochemical factors in Shahu Lake from April 2012 to January 2016 are presented in Table 1. The one-way ANOVA revealed that all physicochemical factors had significant seasonal differences ($P < 0.05$). After spring, WT increased, reaching a maximum (∼29.4 °C) in summer, fell in autumn and dropped to a minimum (∼9.2 °C) in winter. Conductivity had an average range ($\pm$SE) from $90.2 \pm 15.4$ to $532.6 \pm 446.2$ µS/cm with a minimum value of 60.7 µS/cm in April 2013 and a maximum value of 1049 µS/cm in October 2013. DO and water turbidity were highest ($11.0 \pm 1.2$ mg/L, and $142.1 \pm 75.2$ NTU, respectively) in winter and lowest ($5.9 \pm 2.1$ mg/L, and $35.1 \pm 27.4$ NTU, respectively) in summer. In contrast, pH was lowest ($6.7 \pm 0.6$ mg/L) in winter and highest ($7.6 \pm 0.7$ mg/L) in summer.

**Table 1  Mean values (± standard error) of physicochemical factors and their effects on the density of zooplankton in Shahu Lake ($n = 48$).**

|  | April | July | October | January | F | P |
|---|---|---|---|---|---|---|
| Water temperature (°C) | 20.8 ± 1.13[a] | 29.4 ± 0.39[b] | 21.3 ± 0.21[a] | 9.2 ± 0.58[c] | 152.48 | <0.001 |
| Conductivity (μS/cm) | 90.2 ± 4.35[a] | 229.0 ± 68.51[ab] | 532.6 ± 128.79[b] | 279.3 ± 94.28[ab] | 4.52 | 0.008 |
| Dissolved oxygen (mg/L) | 8.8 ± 0.15[a] | 5.9 ± 0.60[b] | 8.9 ± 0.20[a] | 11.0 ± 0.35[c] | 32.74 | <0.001 |
| pH | 7.1 ± 0.20[ab] | 7.5 ± 0.21[a] | 7.1 ± 0.20[ab] | 6.7 ± 0.16[b] | 3.08 | 0.037 |
| Turbidity (NTU) | 82.1 ± 15.37[ab] | 35.1 ± 7.92[a] | 112.4 ± 23.84[b] | 142.1 ± 21.69[b] | 6.24 | 0.001 |

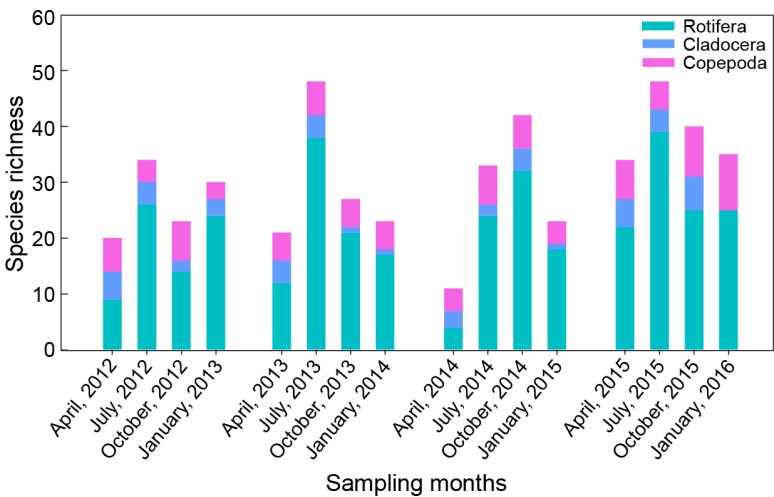

**Figure 2  Seasonal variation in species richness of main zooplankton groups in Shahu Lake from April 2012 to January 2016.**

## Species composition
### Species richness

A total of 115 species of zooplankton were found (Table A1). There were 87 species of Rotifera, 13 species of Cladocera and 15 species of Copepoda. These three main species made up 76.1%, 11.1% and 12.8% of the total species number, respectively. Zooplankton species richness had no significant interannual variation. There were 56 species captured in 2012, 65 species in 2013, 61 species in 2014 and 72 species in 2015 (Fig. 2). Only 24 species appeared simultaneously over the four years: 18 rotifera species, two cladocera species and four copepoda species. Zooplankton species richness exhibited significant seasonal differences ($P = 0.041$). Rotifers, which made up 36.4–81.3% of the total species number, were the dominant component in every season. A total of 58 species was found in spring with the minimum (11 species) in 2014 and the maximum (34 species) in 2015. There were 88 species collected in summer, with the minimum (33 species) in 2014 and the maximum (48 species) in 2013 and 2015. In autumn, 72 species were captured, and the minimum (23 species) were found in 2012 and the maximum (42 species) were found in 2014. In

**Table 2  Dominant species, mean density (ind./L) and dominance (Y) for each year in Shahu Lake during 2012–2015.**

| Dominant species | 2012 ind./L (Y) | 2013 ind./L (Y) | 2014 ind./L (Y) | 2015 ind./L (Y) | Code |
|---|---|---|---|---|---|
| **Rotifera** | | | | | |
| *Brachionus angularis* | 0.9 (0.001) | 12.6 (0.024) | 8 (0.006) | 5.7 (0.023) | S1 |
| *Brachionus forficula* | 5.1 (0.004) | 3.2 (0.004) | 1.1 (0.000) | 5.8 (0.023) | S2 |
| *Brachionus diversicornis* | 7.6 (0.013) | 3 (0.006) | 0.5 (0.000) | 5.6 (0.022) | S3 |
| *Keratella cochlearis* | 35.1 (0.066) | 17.7 (0.052) | 28 (0.030) | 1.8 (0.011) | S4 |
| *Keratella. valga* | 5.8 (0.008) | 16.5 (0.049) | 13.1 (0.009) | 22.7 (0.088) | S5 |
| *Asplanchna priodonta* | 4.8 (0.003) | 12 (0.035) | 25.6 (0.018) | 5.1 (0.034) | S6 |
| *Asplanchna. girodi* | 9.3 (0.021) | 1 (0.001) | 0.1 (0.000) | 0.1 (0.000) | S7 |
| *Asplanchna. brightwel* | 1 (0.000) | 19.6 (0.037) | 1.9 (0.001) | 0.2 (0.000) | S8 |
| *Ascomorpha ecaudis* | – | 30.9 (0.134) | 6.5 (0.003) | 0.2 (0.000) | S9 |
| *Trichocerca cylindrical* | 11.3 (0.013) | 0.1 (0.000) | 12.9 (0.009) | 4.9 (0.020) | S10 |
| *Polyarthra trigla* | – | 4.9 (0.013) | 23.5 (0.017) | 4.2 (0.023) | S11 |
| *Polyarthra dolichoptera* | 57.2 (0.139) | 3.6 (0.007) | 3.9 (0.001) | 2.3 (0.009) | S12 |
| *Polyarthra vnlgaris* | 81.9 (0.133) | – | 10 (0.004) | – | S13 |
| *Synchaeta oblonga* | 24.6 (0.031) | 3.7 (0.006) | 2.4 (0.001) | 0.6 (0.002) | S14 |
| *Filinia longiseta* | 3.4 (0.002) | 0.3 (0.000) | – | 10.3 (0.042) | S15 |
| *Conochilus unicornis* | – | 0.8 (0.001) | 495.6 (0.654) | 22.1 (0.167) | S16 |
| **Cladocera** | | | | | |
| *Diaphanosoma brachyurum* | 2.8 (0.034) | 3.3 (0.009) | 6.5 (0.042) | 8.5 (0.210) | S17 |
| *Bosmina longirostris* | 15.2 (0.267) | 43.3 (0.551) | 66.4 (0.808) | 4.7 (0.160) | S18 |
| *Bosmina. coregoni* | – | 10.9 (0.026) | 8.6 (0.043) | – | S19 |
| *Bosminopsis deitersi* | 1.5 (0.007) | – | – | 5.3 (0.083) | S20 |
| *Daphnia pulex* | 8.4 (0.053) | – | – | – | S21 |
| *Daphnia hyalina* | 5.5 (0.030) | – | – | – | S22 |
| **Copepoda** | | | | | |
| *Copepods nauplii* | 18.6 (0.264) | 48.7 (0.374) | 37.4 (0.479) | 33.1 (0.677) | S23 |
| *Limnoithona sinensis* | – | 13.6 (0.078) | 5.1 (0.038) | 3.1 (0.033) | S24 |
| *Macrocyclops fuscus* | 18.6 (0.198) | – | – | 0.2 (0.001) | S25 |
| *Tropocyclops prasinus* | 6 (0.048) | – | – | – | S26 |
| *Microcyclops varicans* | 12.8 (0.090) | 33.6 (0.161) | 16.6 (0.179) | 4.7 (0.072) | S27 |
| *Mesocyclops leuckarti* | – | – | 4.3 (0.015) | 5.3 (0.066) | S28 |

**Notes.**
 –, the species density is very small or does not appear.

winter, 65 species were identified, and the minimum (23 species) were found in 2012 and the maximum (35 species) were identified in 2015.

### Dominant species

There were 13 dominant species, 10 dominant species, 16 dominant species and eight dominant species in each year from 2012 to 2015 (Table 2). *Bosmina longirostris*, copepod nauplii and *Microcyclops varicans* dominated over the four years. In spring, *Keratella cochlearis* and *Conochilus unicornis* were dominant species. Specifically, spring 2014 saw

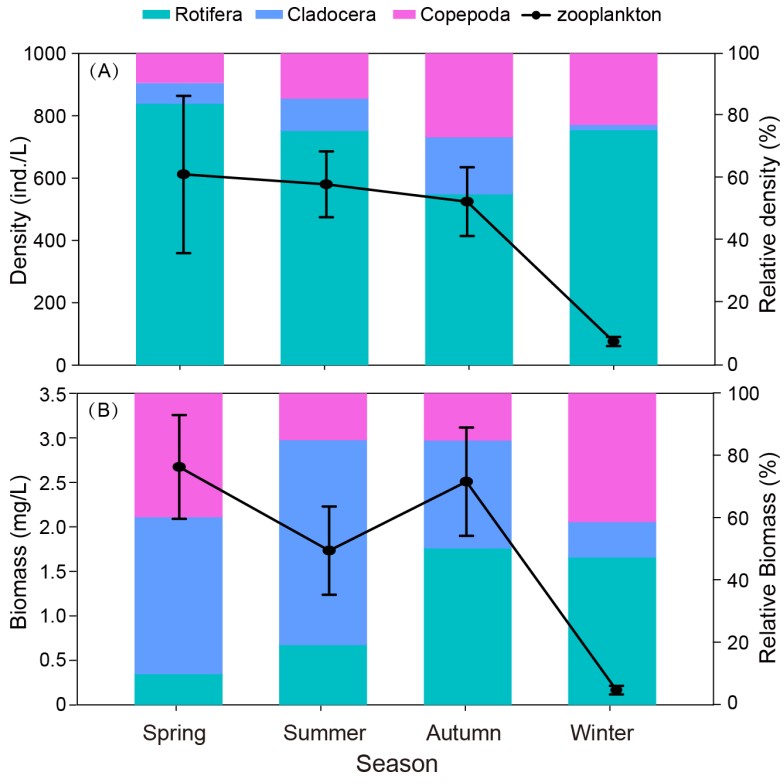

**Figure 3 Seasonal variation in mean density (ind./L) and biomass (mg/L), relative density and biomass (%) of each group (Rotifera, Cladocera and Copepoda) in Shahu Lake during 2012–2015.** (A) Density variation. (B) Biomass variation.

the outbreak of *C. unicornis* which led to the highest density (1908.8 ind./L) of rotifers. In summer and autumn, the dominant genera of rotifers were *Brachionus*, *Keratella*, *Polyarthra*, *Asplanchna* and *Trichocerca*. In winter, the dominant species were replaced by *Polyarthra dolichoptera*, *Synchaeta oblonga*, *K. cochlearis*, *C. unicornis* and *A. priodonta*.

## Zooplankton density and biomass
### Seasonal variation

Total density of zooplankton showed similar trends with species richness (Fig. 3A). In general, maximum density occurred in summer or autumn and minimum density appeared in spring or winter. Zooplankton density was highest in autumn (140.0 ind./L), followed by summer (83.0 ind./L) and spring (56.9 ind./L). The minimum density was found in winter (1.3 ind./L). Rotifer density showed no significant seasonal difference ($P = 0.123$). However, the densities of cladocerans and copepods in winter were significantly lower when compared to the other seasons ($P < 0.001$). The maximum density of cladocerans was observed in October 2014 (219.2 ind./L), while the maximum density of copepods was observed in July 2014 (137.6 ind./L).

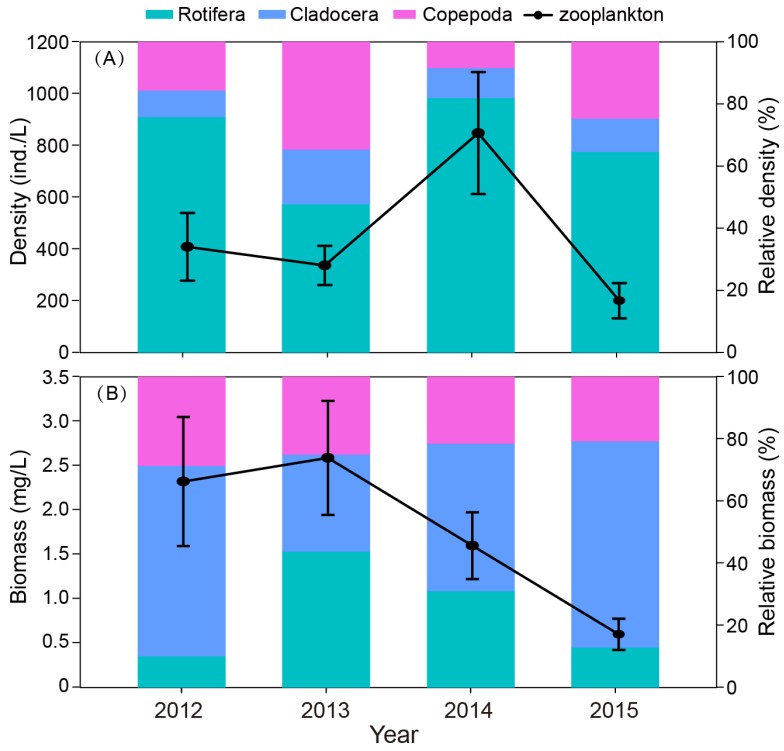

**Figure 4 Interannual differences in density (ind./L) and biomass (mg/L), relative density and biomass (%) of zooplankton in Shahu Lake during 2012–2015.** (A) Density variation. (B) Biomass variation.

The biomass of zooplankton was significantly lower in winter (Fig. 3B) when compared to the other seasons ($P < 0.05$). The highest biomass of rotifers was in autumn, and the lowest was in winter. The biomass of both cladocerans and copepods was highest in spring and lowest in winter. Although the density of cladocerans was lower than those of rotifers and copepods, they contributed 50% of the total biomass of zooplankton and were 1.7 times and 1.9 times the biomass of rotifers and copepods.

### Interannual variation

The interannual variation in zooplankton density was significant ($P = 0.012$). The density in 2014 was significantly higher than in the other three years (Fig. 4A). The outbreak of *C. unicornis* in spring 2014 resulted in the highest density of rotifers ($P = 0.018$). The density of cladocera in 2014 was significantly higher than in 2012 and 2015 ($P = 0.039$). The biomass of zooplankton in 2015 was significantly lower than in the other three years ($P = 0.036$, Fig. 4B). The density and biomass of zooplankton in spring 2015 was very low (18.2 ind./L, 0.16 mg/L, respectively), and the density and biomass of 2015 were lower than in previous years.

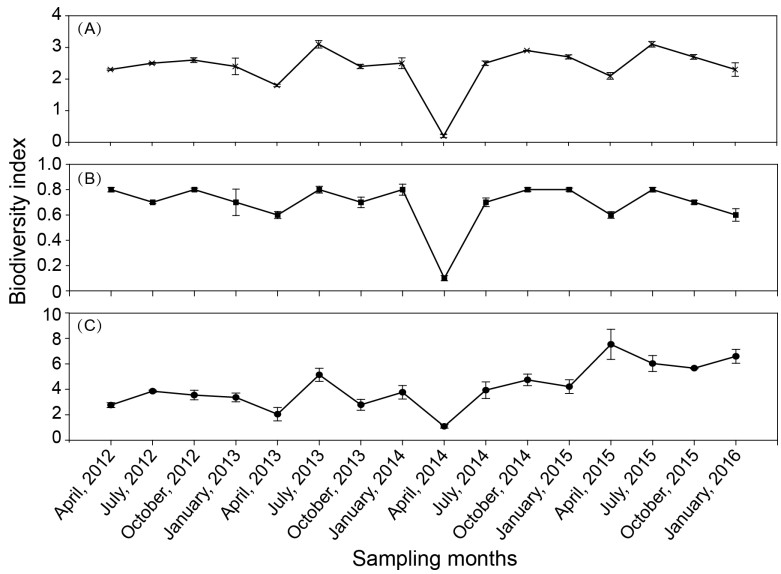

**Figure 5   Seasonal variation in biodiversity index in Shahu Lake during 2012–2015.** (A) Shannon–Weiner index, $H'$. (B) Margalef index, D. (C) Pielou's index, $J'$.

### Species diversity index

There was some fluctuation in the zooplankton diversity index over the 16 seasons (Fig. 5). The Shannon–Weiner index ($H'$) was in the range of 0.2–3.1 (Fig. 5A), with an average of 2.37. The Margalef index (D) was in the range of 1.1–7.6 (Fig. 5B), with an average of 4.2. Pielou's evenness index ($J'$) was in the range of 0.09–0.85 (Fig. 5C), with an average of 0.69. The results of the one-way ANOVA revealed that the Shannon–Wiener index and Pielou's index had significant seasonal variation ($P < 0.001$, and $P = 0.002$, respectively). The seasonal variation in the Margalef index was not significant.

### Community structure

The NMDS results revealed that, apart from July 2012, zooplankton in July and October in all four years were at a high density and had similar dominant species. In addition, these were combined as a summer–autumn community (Fig. 6). Zooplankton in January were categorized as a low-density winter community. The zooplankton community in July 2012 and April in all four years were separated into independent branches because the species composition and density of zooplankton in these seasons were quite different from those in the other seasons. During the month, the species composition and diversity of zooplankton were quite different and formed separate communities. However, the inter-annual zooplankton communities could not be distinguished from each other. This result indicated that the seasonal variation of the zooplankton community structure in Shahu Lake was greater than the interannual variety. In order to identify the key indicator species of the three main NMDS groups (i.e., spring, summer-autumn and winter), we

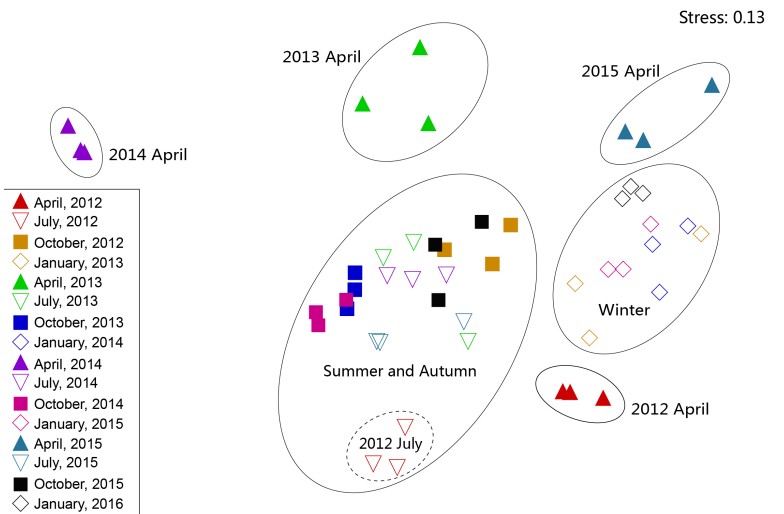

**Figure 6    Non-metric multidimensional scaling ordination (NMDS) of zooplankton communities.**

used the indicator value method (IndVal), and it was revealed that the three groups were characterized by different indicator species (Table A2).

### Redundancy analysis of zooplankton and environmental factors

Before the redundancy analysis (RDA), a preliminary detrended correspondence analysis (DCA) on species-sample data produced the longest gradient length of 3.184, suggesting that both RDA and canonical correspondence analysis (CCA) were appropriate. The RDA was selected to illustrate the relationships between the dominant species of zooplankton and environment factors (Fig. 7). The first axis explained the 15.6% of variance in the species data, and the 50% of variance in the species–environment relationship (Table 3). The second axis explained the 7.4% of variance in species data, and the 23.7% of the variance in the species–environment relationship. The Monte Carlo permutation test revealed that WT ($P = 0.002$), conductivity ($P = 0.002$), pH ($P = 0.018$) and DO concentrations ($P = 0.026$) had significant effects on zooplankton communities. WT had a higher correlation with Axis 1 ($R = 0.695$), and Spearman rank correlation analysis indicated that WT had a significant positive correlation with zooplankton ($R = 0.722$, $P < 0.05$).

## DISCUSSION

### Temporal pattern of zooplankton communities in the sub-lake

Rotifera are an important component of zooplankton communities in a freshwater lake. Their small size, fast growth rate and parthenogenetic reproduction (*Gilbert, 1999*; *Inaotombi, Gupta & Mahanta, 2016*) lead to a generally dominant abundance (*Romo, 1990*). In the present study, rotifers were also the dominant group in Shahu Lake. The quarterly survey from 2012–2015 identified 87 Rotifers, 13 Cladocerans and 15 Copepods, with an average of 63 species each year. However, the species richness was lower when
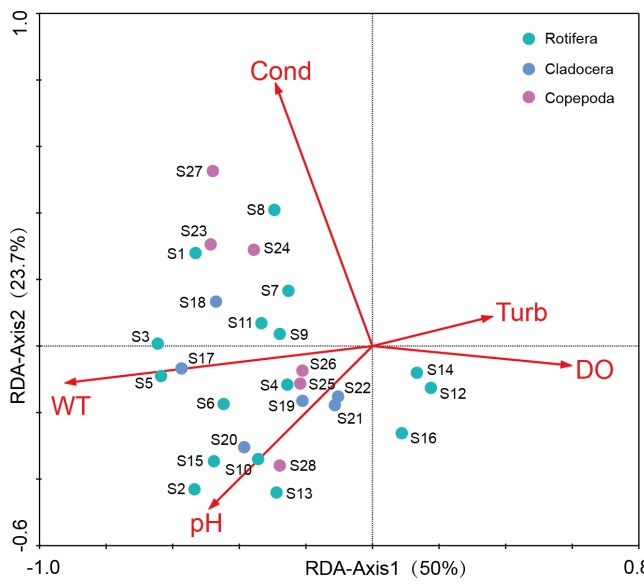

**Figure 7** Redundancy analysis (RDA) of zooplankton dominant species and environmental factors in Shahu Lake (WT, water temperature; Cond, conductivity; DO, dissolved oxygen; Turb, turbidity).

**Table 3  Eigenvalues of the first and second axes in the redundancy analysis.**

| Axes | RDA1 | RDA2 | Total variance |
|---|---|---|---|
| Eigenvalues: | 0.156 | 0.074 | 1 |
| Species-environment correlations: | 0.754 | 0.806 | |
| Cumulative % variance | | | |
|     of species data: | 15.6 | 23 | |
|     of species-environment relation: | 50 | 73.7 | |
| Sum of all eigenvalues | | | 1 |
| Sum of all canonical eigenvalues | | | 0.313 |

compared with the historical research records of Lake Poyang (*Xie, Li & Li, 1997*; *Xie & Li, 1998*; *Wang et al., 2003*; *Huang & Guo, 2007*). The density of three species (*B. longirostris*, Copepod nauplii and *M. varicans*) of zooplankton that were dominant species in four years were separately analyzed, and it was found that the density changes had similar patterns (Fig. 8). In both the flood season (summer) and retreat period (autumn), the density of these three zooplankton were significantly lower in high water level years than in low water level years. The comparative analysis revealed that these changes were closely correlated to the inter-annual hydrological situation (*Gal, Skerjanec & Atanasova, 2014*), and that these might have correlations with the varying number of fishes entering the lake under different water levels. Usually, these three dominant species (all crustaceous zooplankton) are food resources for planktonic feeding fishes (*Mamani, Koncurat & Boveri, 2019*). Hence, the

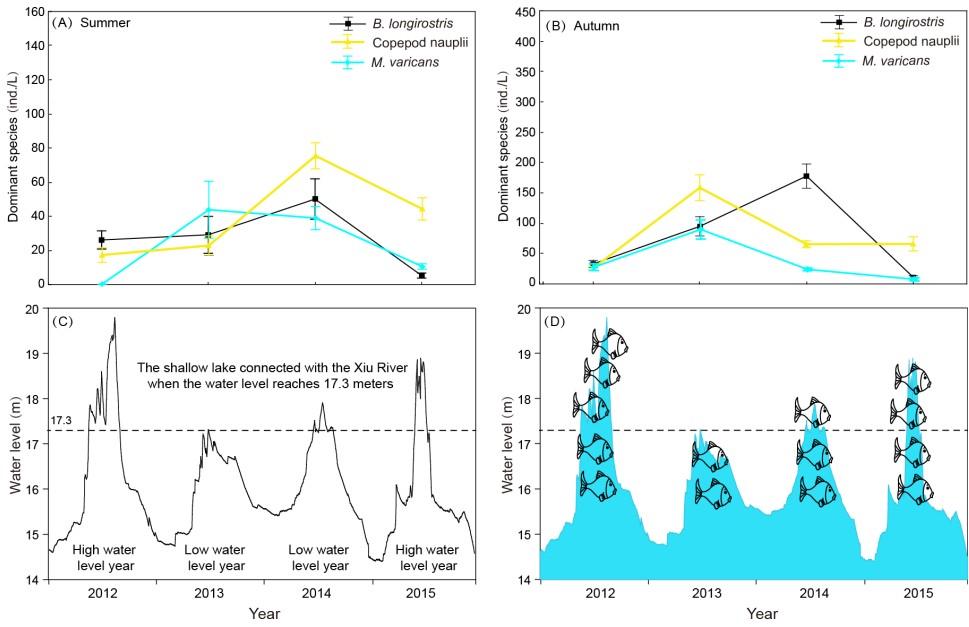

**Figure 8 Interannual variation of zooplankton dominant species density in summer and autumn and diurnal variation of water level in 4 years of Shahu Lake.** (A) Interannual variation of zooplankton dominant species density in summer. (B) Interannual variation of zooplankton dominant species density in autumn. (C) diurnal variation of water level in 4 years. (D) Differences in the number of fishes that may enter the sub-lake after hydrological connectivity.

predation pressures caused by fish might be the direct cause of changes in zooplankton density. The habitat diversity of Lake Poyang is higher than that of Shahu Lake, its sub-lake. Furthermore, the samples in the present study were only taken in the open water area, and the lake has faced intensive human activities, such as beach grazing, fishing, eutrophication caused by pollution, etc. These objective factors may have led to a decrease in species richness. The NMDS analysis suggested that seasonal variation was more significant than interannual in the zooplankton community structure, and that this could be divided into three community groups associated with distinct indicator species (Fig. 6, Table A2). Previous studies of zooplankton in Lake Poyang (*Xie, Li & Li, 1997*; *Xie & Li, 1998*; *Liu et al., 2016*) have roughly observed the seasonal dynamics of the zooplankton community structure. Rotifers peaked in summer and autumn. Cladocerans and copepods achieved their peaks in spring, summer and autumn. However, all three groups were at minimum levels in winter. The present study also showed the same seasonal dynamic patterns. Though the sub-lake was separated from Lake Poyang in the dry season, the seasonal dynamics of the zooplankton community in Shahu Lake were similar to those of Lake Poyang. Similar patterns of seasonal changes in the zooplankton community have been reported in other lakes (*Hu, Yang & Liu, 2014*; *Lin et al., 2014*).

The density and biomass of zooplankton exhibited a significant difference among seasons ($P = 0.035$, $P = 0.002$). Over the four years, rotifers were the main component of zooplankton, representing 72.3% of the total zooplankton abundance with 6.5 times and 4.4 times the density of cladocerans and copepods, respectively. Zooplankton density was highest in autumn and lowest in winter. With one exception, the maximum density (1971.0 ind./L) occurred in spring 2014 due to the outbreak of *C. unicornis*. The biomass of zooplankton was significantly lower in winter than in other seasons ($P < 0.05$). The highest biomass of rotifers was in autumn and lowest in winter. An earlier study reported that cladocerans and copepods are the main component of zooplankton productivity due to their larger body size (*Castro & Gonçalves, 2007*). In the present study, we also found that the biomass of both cladocerans and copepods was highest in spring. Although the density of cladocerans was lower than rotifers and copepods, this contributed to 50% of the total biomass of zooplankton.

We found that the seasonal succession characteristics of the zooplankton community in Shahu Lake were consistent with a previously reported model (*Sommer, 1986*). In winter, the cold temperature and lack of food resulted in a decline in zooplankton reproductive capacity. Thus, a minimum zooplankton density was observed in this period. In spring, the phytoplankton biomass increased with rising temperatures, and they provided a greater food resource to phytoplanktivorous zooplankton (Cladocera and Calanoida). Simultaneously, the hatching of dormant eggs and copepods diapause ontogeny developed into supplementary populations. The result was an increase in zooplankton abundance in spring (*Hairston, Hansen & Schaffner, 2000*). The numbers of *Daphnia* gradually decreased after midsummer, and this was replaced by smaller species and copepods (*Threlkeld, 1979*; *Steiner, 2004*; *Deng et al., 2008*). After autumn, with fishing making fish predation less of a threat, the abundance of rotifers rapidly increases, and they become the dominant groups in Shahu Lake.

Some studies have found that the spring-summer zooplankton community is not in a complete repetitive succession in small sub-lakes due to the difference in interannual water temperature and rainfall (*Rettig, Schuman & Mccloskey, 2006*). There was a large variation in the spring zooplankton community of the Shahu Lake during the four years, while in the other seasons the community structures tended to be similar. In early spring, Shahu Lake and Lake Poyang were still not connected. Zooplankton communities in Shahu Lake were mainly affected by rainfall, human disturbance and other unspecified factors. Therefore, zooplankton community succession in this period may not have a uniform direction. In summer, however, Shahu Lake was connected with the main lake. The material and biological exchanges between the sub-lake and main lake resulted in a similarity in water environment and biological community structure. Therefore, the zooplankton community succession was back to the early stages (*Baranyi et al., 2002*).

## Effects of environmental factors on the zooplankton community

Water physicochemical factors can affect species composition and the abundance of a zooplankton community. The significant differences in physicochemical factors in different seasons leads to seasonal zooplankton dynamics (*Deyzel, 2004*). Some studies have pointed out that the seasonal dynamics of zooplankton can be influenced by temperature (*Hu, Yang & Liu, 2014*; *Hussain et al., 2016*). Water temperature has an important effect on dormant eggs hatching, growth and the reproduction of zooplankton (*Korpelainen, 1986*; *Hu, Xi & Tao, 2008*). For example, the net reproduction rate of *Brachionus diversicornis* is highest when the temperature is 30 °C (*Ning et al., 2013*), which might be the main reason why *B. diversicornis* is the dominant species in summer in Shahu Lake. Temperature also affects phytoplankton. High temperatures were favourable for the growth of phytoplankton, and the biomass of phytoplankton in Lake Poyang was highest in summer (*Wu et al., 2013*). Low temperatures limit the predation of zooplankton on phytoplankton (*Zheng et al., 2015*). Hence, zooplankton in Shahu Lake have a high density in summer and a low density in winter.

Different zooplankton species have different adaptations to temperature (*Tao, Xi & Hu, 2008*). The number of resting eggs increases in both higher and lower temperatures (*Shi & Shi, 1996*). In the present study, it was found that the dominant species in summer were thermophilic species, such as *Brachionus* spp. and *Trichocerca* spp., and wide suitable temperature species, such as *Keratella* spp. The dominant species in winter were those suited for low temperatures, such as *Polyarthra dolichoptera*, *Synchacta* spp. and so on. Therefore, the seasonal variation of temperature is one of the reasons for the changing zooplankton dominant species. WT variation was significant in Shahu Lake, and was highest in summer and lowest in winter (Table 1). The RDA suggested that there was a positive correlation between temperature and most of the dominant species. The Spearman rank correlation analysis also revealed that temperature has a positive correlation with species richness ($R = 0.376$, $P = 0.009$), density ($R = 0.401$, $P = 0.005$) and biomass ($R = 0.480$, $P = 0.001$) of zooplankton.

The results of the redundancy analysis revealed that conductivity, pH and dissolved oxygen also had a significant effect on the seasonal variation of the zooplankton community. *Bērziņš & Pejler (1987)* reported that some species of rotifers, which could instruct the water oligotrophic conditions, generally appeared in water at pH 7.0 or slightly lower. Some other species of Rotifera indicated that eutrophic conditions prefer water with a pH value higher than 7.0. The pH value of Shahu Lake was higher than 7.0, and its water was at a certain degree of eutrophication. Among its dominant species, such as *Brachionus* spp., *A. brightwelli*, *S. oblonga*, *Filinia longiseta*, *Daphnia pulex*, *Bosmina longirostris* and *Bosmina coregoni*, most were commonly found to be indicator species of eutrophication. Phytoplankton blooms can lead to higher water pH values. There is a correlation between phytoplankton and the water pH value in summer. In the present study, the Spearman rank correlation analysis revealed that significant positive correlations existed between pH, the zooplankton species richness ($R = 0.644$, $P < 0.001$) and the ShannonWeiner diversity index ($R = 0.487$, $P < 0.001$). In the present study, we found that there was a significant positive correlation between conductivity and copepods ($R = 0.463$, $P < 0.001$), but

there was a weakly positive correlation between conductivity and cladocerans ($R = 0.078$, $P \leq 0.597$). This was consistent with a previous study (*Soto & Rios, 2006*).

Water level fluctuation was also one of the important factors that affected the zooplankton community structure. It was found that the density and community structure of zooplankton changed as the water level fluctuated (*Goździejewska et al., 2016*). As the fluctuation intensified, the former dominant species, *Daphnia*, was replaced by rotifers (*Zhou et al., 2016*). The zooplankton composition of Shahu Lake in summer was dominated by small individual rotifers, copepod nauplii and *Bosmina longirostris*. The main reason for this was that Lake Poyang was in the rising water level period from April to July, and the water level changes resulted in a disturbance to zooplankton. When the water level rose, Shahu Lake was connected with the main lake (Fig. 8C). Consequently, nutrients and other biological communities (such as fishes, Fig. 8D) poured into the sub-lake along with the flood, and interactions occurred among zooplanktons and other aquatic organisms from rivers. This was probably one of the reasons for the great shifts in the zooplankton community in Shahu Lake from spring to summer. Interval water level differences can also lead to annual zooplankton differences. In the summer of 2012, the water level was significantly higher than in previous years (Fig. 8C). The continuing high-water level could be the reason why the zooplankton community structure in summer 2012 was significantly different than in other years.

Evaporation, seepage flow and the opening of water-gates for fishing from the middle of October resulted in the water level gradually decreasing in Shahu Lake. The water depth was only 20-30 cm by the end of fishing, exposing most of the lake basin. The lake's bottom sediment and its attachments fully contacted with the atmosphere and the sun. The digestion of organic matter in the sediment was accelerated and the soil structure improved (*Hu, 2012*). However, the water-gate was not opened during the winter of 2013. Hence, the water depth remained more than one meter during that period (Fig. 8C). The stability of the water level, coupled with nutrient enrichment and temperature recovery in spring, maintained a relatively stable environment, leading to the outbreak of the *Conochilus unicornis* population.

## Effects of aquatic organisms on the zooplankton community

In addition to environmental factors, biological factors are also important in changing zooplankton community seasonal dynamics (*Castro & Gonçalves, 2007*). Fish have a choice during predation (*Dodson, 1970*) and most fish prefer bigger zooplankton (*Wang, 2010*). Filter-feeding fishes such as silver carp (*Hypophthalmichthys molitrix*) and bighead carp (*Aristichthys nobilis*) have an important place in Shahu Lake (*Zeng et al., 2015*). After the lake has been enclosed and fished in winter, zooplankton face lower predation pressure from fish at the start of spring. When the water level rises, the floods not only change the zooplankton community structure, but also bring many migrating fishes from the rivers and other lakes. These two factors have led to the miniaturization of zooplankton species. The larger zooplankton, *Daphnia hyalina*, *D. pulex* and *Sinocalanus dorrii*, were dominant in spring. However, the abundance of these species declined sharply in summer, and some species even disappeared from the lake. This suggested a close correlation with

fish predation (*Scheffer et al., 1997*; *Steiner, 2004*; *Deng et al., 2008*), and the outbreak of small C. unicornis in spring 2014 may have been related to the absence of Daphnia caused by the end of fishing in winter 2013.

In addition to predation relations between fish and zooplankton, some other aquatic organisms have contributed to zooplankton seasonal dynamics by affecting the water environment. In winter, the grasslands, mudflats and shallow waters provide an excellent habitat for wintering migratory birds, and a large number of migratory birds live in the Lake Poyang. The feces of winter migratory birds lead to an increase of nitrogen and phosphorus concentrations, which increase the eutrophication level of the sub-lakes. The study of the water quality of Shahu through the zooplankton diversity index revealed that spring water quality was worse than that of other seasons (*Zhu, Liu & Jin, 2014*; *Nie et al., 2018*). The dynamics of the zooplankton community are ecologically complex, and some factors have not been involved in this experiment. The composition and biomass of phytoplankton, interspecific and intraspecific competition, and nutrient concentration all have an effect on the succession of the zooplankton community.

## CONCLUSION

The community structure of zooplankton has a significant seasonal pattern and no interannual repeatability. The differences in zooplankton density, biomass and diversity indices were significant in different seasons and years. This study will be helpful in the further understanding of the ecosystem stability of lakes connected with rivers, and in providing scientific guidance for the protection of lake wetlands.

Overall, ecological environmental protection is very important for the decisions made by the current Chinese government. Promoting green development and strengthening ecological system protections are imperative. As the largest lake in China, Lake Poyang's ecological states are of great importance for the whole Yangtze catchment, and it is a vital part of China's ecological environmental protection including biodiversity conservation, and water resource planning and management. The results of the present study can thereby provide vital scientific basis for lake ecosystem protection and for the sustainable utilization of biodiversity resources.

## ACKNOWLEDGEMENTS

We are grateful to Liu GH, Yan JY, Zeng ZG, Zhang XC, Zeng T, AN CT, Guo GY, Huang WG, Lv Q, Hou JJ, Zhang XL for their assistance in the field and laboratory. We are also grateful to Waigen Huang for improving the language of the manuscript.

## APPENDIX

Hu et al. (2019), *PeerJ*, DOI 10.7717/peerj.7590

**Table A1  Species list of zooplankton in Shahu Lake, 2012–2015.**

| Zooplankton species | 2012 | | | | 2013 | | | | 2014 | | | | 2015 | | | |
|---|---|---|---|---|---|---|---|---|---|---|---|---|---|---|---|---|
| | Spring | Summer | Autumn | Winter | Spring | Summer | Autumn | Winter | Spring | Summer | Autumn | Winter | Spring | Summer | Autumn | Winter |
| **Rotifera** | | | | | | | | | | | | | | | | |
| Anarthra aptera | | | | | | | | | | + | | | | | | |
| Argonotholca foliacea | | | | | | | | | | | | | | ++ | | |
| Ascomorpha ecaudis | | | | | +++ | +++ | +++ | | | + | +++ | | | + | | + |
| Ascomorpha ovalis | | | | | | | + | | | + | ++ | | + | + | + | + |
| Ascomorpha saltans | | +++ | | | + | + | | + | | | + | | | + | | |
| Asplanchna brightwel | | + | | + | | + | +++ | | ++ | + | | | | + | | |
| Asplanchna girodi | | +++ | +++ | +++ | | + | + | + | | | | + | | + | | |
| Asplanchna priodonta | | ++ | | | + | +++ | +++ | | | | +++ | +++ | +++ | +++ | ++ | +++ |
| Asplanchna sieboldi | | + | | | | | | | | | | | | | | |
| Brachionus angularis | + | | ++ | | | ++ | +++ | | | +++ | ++ | | + | +++ | ++ | + |
| Brachionus budapestiensis | + | | +++ | + | + | | + | +++ | | + | ++ | +++ | | +++ | + | |
| Brachionus calyciflorus | + | + | ++ | + | + | | | | | | + | | | + | | +++ |
| Brachionus capsuliflorus | | | | | | | | | | | + | +++ | + | + | + | |
| Brachionus caudatus | | | | | | + | | | | | | +++ | | | | |
| Brachionus diversicornis | + | ++ | +++ | + | + | + | + | + | | | + | + | + | +++ | ++ | |
| Brachionus falcatus | | +++ | | | | | | | | +++ | | +++ | +++ | | | |
| Brachionus forficula | | ++ | | | | + | | + | | | + | ++ | | +++ | +++ | |
| Brachionus leydigi | | | | | | | | | | | | | + | | | + |
| Brachionus urceus | | | | + | | + | | + | | | | ++ | +++ | ++ | + | +++ |
| Cephalodella catellina | | | | | | | | + | | | | | | | | |
| Cephalodella gibba | + | | | + | | | | | | | | | | | | |
| Cephalodella sterea | | | | | | | | | | | | | | + | | |
| Collotheea mutabilis | | | | | | + | | | + | | | | | | | |
| Conochiloides dossuarius | | | | | | | | | | ++ | ++ | | | | | |
| Conochilus unicornis | | | | | | | +++ | +++ | ++ | +++ | +++ | +++ | +++ | +++ | +++ | +++ |
| Eosphora thoa | | | | | | + | + | | | | | | | | | + |
| Eothinia elongata | | | | | | + | | | | | | | | | | |
| Epiphanes senla | | ++ | | | | | + | | | | | | | | | |
| Euchlanis dilatata | | | | | | | | | | | + | | + | + | | |
| Filinia longiseta | | + | | | | + | | | | | | | + | +++ | +++ | |
| Filinia maior | | + | | | | | + | | | | +++ | | | | | |
| Filinia passa | | + | +++ | | | + | + | | | | | | | +++ | + | |
| Gastropus hyplopus | | ++ | + | | + | + | | + | | | | | | | | |
| Gastropus stylifer | | + | + | + | | | | | | + | | | | | | |

Hu et al. (2019), *PeerJ*, DOI 10.7717/peerj.7590

**Table A1** (*continued*)

| Zooplankton species | 2012 | | | | 2013 | | | | 2014 | | | | 2015 | | | |
|---|---|---|---|---|---|---|---|---|---|---|---|---|---|---|---|---|
| | Spring | Summer | Autumn | Winter | Spring | Summer | Autumn | Winter | Spring | Summer | Autumn | Winter | Spring | Summer | Autumn | Winter |
| *Harringia eupoda* | | | | + | | | + | | | | | | | | | |
| *Kellicottia longispina* | | + | | | | | | | | | | | | | | |
| *Keratella cochlearis* | +++ | +++ | | + | | + | +++ | +++ | | ++ | +++ | +++ | ++ | ++ | ++ | + |
| *Keratella quadrata* | | | | | | | + | | | + | | + | | | + | |
| *Keratella ticinensis* | | | | | | | | | | | | | | + | ++ | |
| *Keratella valga* | | ++ | + | | | +++ | +++ | + | | + | +++ | ++ | + | +++ | +++ | |
| *Lecane luna* | | | | | | + | | | | | | | | | | |
| *Lecane nodosa* | | | | | | | | | | | | | | | | + |
| *Lecane ungulata* | | | | | | | | | | | | | | | | + |
| *Lepadella apsida* | | | | + | | | | | | | | | | | | |
| *Lindia truncata* | | | | | | | | | | | + | | | | | |
| *Monostyla crenata* | | | | | | | + | | | | | | | | | |
| *Monostyla elachis* | | | | | | | | | | | | | | ++ | | |
| *Monostyla lunaris* | | | | | | | | | | | | | + | + | | |
| *Monostyla unguitata* | | | | | | | | | | + | + | | | | | |
| *Mytilina ventralis* | | | | | | + | | | | | | +++ | | | | |
| *Notholca labis* | | | | | | | | + | | | | | | | | + |
| *Notommata tripus* | | | | | | + | | | | | | | | | | |
| *Pedalia mira* | | | | + | | + | | | | + | ++ | | | | | |
| *Ploesoma hudsoni* | | | ++ | | + | + | | | + | ++ | | | +++ | | | |
| *Ploesoma truncatum* | | | | + | | + | | | | + | | | | | | |
| *Polyarthra dolichoptera* | +++ | +++ | ++ | +++ | | + | + | +++ | | | | +++ | ++ | + | + | + |
| *Polyarthra euryptera* | | | | | | | | | | ++ | + | | + | | | |
| *Polyarthra trigla* | | | | | + | + | +++ | + | | +++ | +++ | | + | +++ | +++ | +++ |
| *Polyarthra vnlgaris* | ++ | +++ | + | + | | | | | | | +++ | | + | | | |
| *Pompholyx complanata* | | | | + | | | + | | | | | | | | | |
| *Pompholyx sulcata* | | | | | | | | | | + | + | | | | | |
| *Proales daphnicola* | | | | | + | | | | | | | | | | | |
| *Pseudoharringia semilis* | | | | | | + | | | | | | | | | | |
| *Resticula gelida* | | | | | | + | | | | | | | | | | |
| *Resticula melandocus* | | | | | | + | | | | | | | | | | |
| *Scaridum longicaudum* | | ++ | | | ++ | + | | | | | | | | + | | |
| *Synchacta atylata* | | | | + | | | | | | | + | | | | | |
| *Synchacta tremula* | | | | + | | | + | | | | | + | | | | |
| *Synchaeta oblonga* | | +++ | | +++ | + | + | | +++ | | | | +++ | + | | | +++ |
| *Synchaeta pectinata* | | | | | | | | | | | | | | + | + | +++ |
| *Trichocerca bicristata* | | | | | + | + | | | | | + | | | | | |

Hu et al. (2019), *PeerJ*, DOI 10.7717/peerj.7590

**Table A1** (*continued*)

| Zooplankton species | 2012 | | | | 2013 | | | | 2014 | | | | 2015 | | | |
|---|---|---|---|---|---|---|---|---|---|---|---|---|---|---|---|---|
| | Spring | Summer | Autumn | Winter | Spring | Summer | Autumn | Winter | Spring | Summer | Autumn | Winter | Spring | Summer | Autumn | Winter |
| *Trichocerca bicuspes* | | | | | | | | | | | | | | + | + | |
| *Trichocerca capucina* | | + | +++ | + | | +++ | | | | + | +++ | | | ++ | + | |
| *Trichocerca cylindrical* | + | +++ | | + | + | | | | | +++ | +++ | | | ++ | +++ | + |
| *Trichocerca dixon-nuttalli* | | | | | | | | | | | | | | + | | |
| *Trichocerca elongata* | | | | | | | | | | | + | | | + | + | |
| *Trichocerca gracilis* | | | | | | + | | | | | | | | +++ | + | + |
| *Trichocerca longiseta* | | ++ | | + | | | | | | | | + | | | | |
| *Trichocerca lophoessa* | | ++ | + | | | + | | | | | + | | | ++ | + | + |
| *Trichocerca pusilla* | | | | | | | | | | | ++ | | | ++ | + | + |
| *Trichocerca rattus* | | | | | | | | | | | | | | + | | + |
| *Trichocerca rousseleti* | | | | | | | | | | | + | | | | | |
| *Trichocerca similis* | | | | | | | | | | | | | + | + | + | |
| *Trichocerca stylata* | | | | | | + | | | | + | +++ | | | | | |
| *Trichocerca tenuior* | | | | | | | | | | + | | | | + | | |
| *Trichocerca weberi* | | | | + | | + | + | | | | | | | | | + |
| *Trichotria tetractis* | | | | | | | | | | | | | + | | | + |
| **Cladocera** | | | | | | | | | | | | | | | | |
| *Alonella rostrata* | | + | | | | | | | | | | | | | | |
| *Bosmina coregoni* | | | | | | +++ | | | ++ | | +++ | | | | | |
| *Bosmina fatalis* | | | | | | | | | | | | | + | | ++ | |
| *Bosmina longirostris* | + | +++ | +++ | + | +++ | + | +++ | + | ++ | +++ | +++ | ++ | +++ | ++ | +++ | |
| *Bosminopsis deitersi* | | + | | + | | | | | | | | | | +++ | + | |
| *Daphnia cucullata* | | | | + | | | | | | | | | +++ | | | |
| *Daphnia hyalina* | +++ | | | | | | | | | | | | | | | |
| *Daphnia pulex* | +++ | | | | | | | | | | | | + | | | |
| *Diaphanosoma brachyurum* | + | + | +++ | | | +++ | | | + | ++ | ++ | | + | +++ | +++ | |
| *Diaphanosoma leuchtenbergianum* | | | | | + | + | | | | | | | | + | + | |
| *Leptodora kindti* | | + | | | + | | | | | | | | | | | |
| *Moina micrura* | | | | | + | | | | | | | | | | | |
| *Sida crystallina* | + | | | | | | | | | | ++ | | | | ++ | |
| **Copepoda** | | | | | | | | | | | | | | | | |
| *Copepod nauplii* | +++ | ++ | +++ | +++ | ++ | +++ | +++ | +++ | + | +++ | +++ | +++ | +++ | +++ | +++ | +++ |
| *Cyclops vicinus* | | | | | | | | | | | | | + | | | + |
| *Limnocletodes behningi* | | | | | + | + | | | | | | | | | | |
| *Limnoithona sinensis* | | | | | | +++ | + | +++ | + | +++ | + | | | ++ | +++ | + |

**Table A1** (*continued*)

| Zooplankton species | 2012 | | | | 2013 | | | | 2014 | | | | 2015 | | | |
|---|---|---|---|---|---|---|---|---|---|---|---|---|---|---|---|---|
| | Spring | Summer | Autumn | Winter | Spring | Summer | Autumn | Winter | Spring | Summer | Autumn | Winter | Spring | Summer | Autumn | Winter |
| *Macrocyclops fuscus* | +++ | ++ | ++ | | | | | | | | | | +++ | | | |
| *Mesocyclops leuckarti* | | | | | | | | | | ++ | | | | +++ | +++ | + |
| *Microcyclops varicans* | +++ | | +++ | +++ | | + | +++ | + | | +++ | +++ | +++ | + | +++ | +++ | ++ |
| *Neodiaptomus schmackeri* | | | | | | | | | | | | | ++ | | ++ | + |
| *Paracyclops fimbriatus* | | | | | | | | | | | | | + | | | + |
| *Schmackeria forbesi* | + | | + | | | | | | ++ | ++ | + | | | | + | + |
| *Sinocalanus dorrii* | ++ | + | + | | +++ | + | + | + | ++ | + | | + | ++ | | + | + |
| *Thermocyclops hyalinus* | | | | + | | | | | | + | | | | | | |
| *Thermocyclops kawamurai* | +++ | | + | | ++ | + | + | + | | ++ | + | + | | | + | + |
| *Thermocyclops taihokuensis* | | | | | | | | | | | | | | + | ++ | |
| *Tropocyclops prasinus* | +++ | + | ++ | | | | | | | | | | | | | |

**Notes.**

Note: + means appeared; ++ means common species (occurrence frequency greater than 0.65); +++ means dominant species (dominance index greater than 0.02).
**Table A2** Summary of indicator species analysis showing indicator value (IV) and *p* values for each group.

| | Group | IV | P values |
|---|---|---|---|
| *Sinocalanus dorrii* | S | 90.62 | 0.001 |
| *Daphnia pulex* | S | 33.33 | 0.004 |
| *Macrocyclops fuscus* | S | 45.22 | 0.012 |
| *Daphnia hyalina* | S | 25.00 | 0.032 |
| *Brachionus angularis* | SA | 86.71 | 0.001 |
| *Brachionus forficula* | SA | 56.50 | 0.001 |
| *Brachionus diversicornis* | SA | 73.05 | 0.001 |
| *Keratella valga* | SA | 90.29 | 0.001 |
| *Asplanchna priodonta* | SA | 68.48 | 0.001 |
| *Ascomorpha ovalis* | SA | 52.65 | 0.001 |
| *Trichocerca cylindrical* | SA | 58.97 | 0.001 |
| *Trichocerca capucina* | SA | 63.77 | 0.001 |
| *Pedalia mira* | SA | 56.78 | 0.001 |
| *Diaphanosoma brachyurum* | SA | 72.27 | 0.001 |
| *Bosmina longirostris* | SA | 69.57 | 0.001 |
| Copepod nauplii | SA | 75.76 | 0.001 |
| *Limnoithona sinensis* | SA | 63.97 | 0.001 |
| *Microcyclops varicans* | SA | 71.50 | 0.001 |
| *Polyarthra trigla* | SA | 72.21 | 0.002 |
| *Filinia longiseta* | SA | 41.64 | 0.007 |
| *Keratella cochlearis* | SA | 64.85 | 0.009 |
| *Mesocyclops leuckarti* | SA | 37.17 | 0.009 |
| *Collotheca mutabilis* | SA | 33.33 | 0.014 |
| *Brachionus falcatus* | SA | 34.57 | 0.019 |
| *Asplanchna brightwel* | SA | 46.87 | 0.021 |
| *Trichocerca stylata* | SA | 29.17 | 0.022 |
| *Scaridum longicaudum* | SA | 31.97 | 0.023 |
| *Filinia maior* | SA | 29.17 | 0.027 |
| *Bosminopsis deitersi* | SA | 28.12 | 0.039 |
| *Brachionus budapestiensis* | SA | 44.18 | 0.05 |
| *Synchaeta oblonga* | W | 62.10 | 0.001 |

**Notes.**

S, spring; SA, summer and autumn; W, winter.

### Funding

This project was supported by the National Natural Science Foundation of China (31560133, 41501028), the International Crane Foundation (11001903) and the Key Laboratory of Ministry of Education, Nanchang University (13006457). The funders had

no role in study design, data collection and analysis, decision to publish, or preparation of the manuscript.

### Grant Disclosures
The following grant information was disclosed by the authors:
National Natural Science Foundation of China: 31560133, 41501028.
International Crane Foundation: 11001903.
Key Laboratory of Ministry of Education, Nanchang University: 13006457.

### Competing Interests
The authors declare there are no competing interests.

### Author Contributions
- Beijuan Hu analyzed the data, authored or reviewed drafts of the paper, approved the final draft.
- Xuren Hu performed the experiments, analyzed the data, prepared figures and/or tables, authored or reviewed drafts of the paper.
- Xue Nie performed the experiments.
- Xiaoke Zhang authored or reviewed drafts of the paper.
- Naicheng Wu analyzed the data, prepared figures and/or tables, authored or reviewed drafts of the paper.
- Yijiang Hong contributed reagents/materials/analysis tools.
- Hai Ming Qin conceived and designed the experiments, performed the experiments, analyzed the data, prepared figures and/or tables, authored or reviewed drafts of the paper, approved the final draft.

### Data Availability
The raw data of zooplankton density and biomass, and synchronously determined physicochemical factors, are available in the Supplemental Files.

### Supplemental Information
Supplemental information for this article can be found online at http://dx.doi.org/10.7717/peerj.7590#supplemental-information.

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
