# Peer review of "Seasonal and inter-annual community structure characteristics of zooplankton driven by water environment factors in a sub-lake of Lake Poyang, China"

_PeerJ, doi:10.7717/peerj.7590_

## Round 0.1 · original submission · Major Revisions

The manuscript could contribute to our understanding of the temporal variations and underlying drivers of zooplankton community in Lake Poyang. The reviewers, however, raised several substantial comments in how improving the quality and clarity of the draft. These include the English writing, phase definition, and the supporting data analyses for key findings. Please submit a list of changes or a rebuttal against each point raised by the reviewers.

Reviewer 1 ·

Basic reporting

The article has high quality figures and tables, but the language still need some improvement. Besides, I would recommend the authors add the relevant references to the methods which have used in the article.

Experimental design

The research question has well defined on a meaningful topic. However, I have worried about the method for rotifer collection. The 64 um plankton net could be too large to loss a high mount of rotifer species, density as well as biomass for the zooplankton community.

Validity of the findings

There is a sufficient data base. However, the topic mentioned "different hydrological years", while I didn't find sufficient description on this part. I would suggest either change the topic or add related information to support the hydrological differences and influences.

Additional comments

1, Change "shallow lake" to "sub-lake" in the abstract. and explain "shallow lake" in the introduction when first using it. Maybe you can add "linking" or "surrounding" to identify the shallow lake. Otherwise readers will get confused when first look at it.
2, Line 65: delete "water".
3, line 71: "biological" change to "biotic".
4, line 72: "small algae" is not professional expression.
5, line 77: top-down "control"?
6, line 85: delete "characteristics of".
7, line 86: "repeated water changes" looks strange. Maybe you mean "fluctuation of water level"?
8, line 97: "remain rarely reported" has grammar mistake.
9, line 101-103: "zooplankton" is a collective noun following with plural form.
10, line 103: "water" change to "aquatic".
11, line 104: better divided into two sentences.
12, line 150: As I mentioned above, each method should link to a reference.
13, line 282-287: There are actually your results. In this part, you made too many duplication of your results, but lack of powerful and meaningful discussion to extend your results. I would suggest you think about the dominant species, and the species who can survive in the winter. How about they biological traits? Are there any relative reports of them in other place? Moreover, the diversity patterns are also lack of discussions.
14, line 435-437: This conclusion neither with results to support nor discussion on it.

Reviewer 2 ·

Basic reporting

The article in English use clear, unambiguous, technically correct. The article professional standards of courtesy and expression.

Experimental design

All variables were ln (x+1) transformed prior to analysis?

Does this have any references?

Is it reasonable?

Validity of the findings

good

Reviewer 3 ·

Basic reporting

The authors used the correct scientific definitions however, the linguistic quality of the manuscript is very low. The manuscript needs a revision by an English native speaker since, in many parts, it is not clear what do authors want to say.
The title is not very relevant to the study due to the inclusion of the phrase “different hydrological years”. Since the aim of the study is the investigation of the seasonal and annual changes in zooplankton community structure and no hydrological analyses are included in the manuscript I strongly recommend that the above phrase is removed from the title.
Literature and background information: In my opinion, literature and background information is satisfactory. The literature is relevant to the study area and the content of the study in general. However, I have a comment about the introduction since it needs to be more focused on seasonal changes of species presence and abundance as well as community structure, rather than on the effects of eutrophication since water quality parameters were not measured in the present study.
Structure for their manuscript: The authors followed the required structure for their manuscript. They have provided good quality figures, Tables and raw data. However, I have a comment about one of the figures provided. Particularly, Figure 1 caption is incomplete “Location of Shahu Lake and the zooplankton sampling points (water depth map based on the water level of October 2012)”. The authors should name each one of the individual maps shown in Fig. 1 (i.e. a, b, c, d) and include information about each one of them to the caption. It would be also important to give numbers to the sampling points.
References: I have noticed that some of the references in the text are not included in the Reference section i.e. Hu, 2015; Lin, 2013; Yang & Huang, 1994. I strongly recommend careful checking of the references provided.

Experimental design

The scientific gap is well documented and research questions are stated clearly. However, the last paragraph of the introduction includes some points which need improvement and clarification, i.e.
- The authors stated in line 109: “This study has carried a preliminary research into seasonal variations in zooplankton communities in Shahu Lake, a sub-lake of Lake Poyang.” What do the authors mean by the term preliminary? Why the results are preliminary?
- My major objection concerns research question 2 Line 112: “(2) identify the dominant environmental factors that affect the variation in zooplankton communities.” the term “environmental factors” is very generic, and definitely it includes more factors that are known to affect freshwater biota i.e. depth and more importantly nutrients concentration. And this is my major concern about this study. I recommend that the authors reconsider their research question 2 and if they can answer this question by only testing water temperature, conductivity, DO, pH and turbidity.
Sampling design of zooplankton is described adequately.
In my opinion, the authors should clarify the criteria for selecting the three points. They mentioned in the text that “With the water level declining, the water only remained in the deepest area, so three sampling points were set in the more than 1.4 m area in winter (Fig. 1).” So, what I understood is that all sampling points are located in areas with a water level being at 1.4m depth at least. However, according to Fig 1 this is not true. 2. Also, Fig. 1 caption needs more elaboration (see paragraph 1).
Data analysis
Subtitles in the Data analysis section is necessary in order to be easier for the reader to follow your statistical analysis. i.e Zooplankton community structure
I also believe that density and biomass estimation should be moved in this section since is also part of community structure analysis. Their density was calculated by dividing the individual numbers of zooplankton gathered in each sample by the sample volume and expressed by ind./L. The biomass of zooplankton (wet weight) was evaluated according to the method of Zhang and Huang (1991). The weight of each nauplii was estimated to be about 0.003 mg (Xie & Li, 1998).

Validity of the findings

I have a major objection about the conclusion that the authors ended up and it is also stated in the abstract “Water environmental factors, water level fluctuations, wintering migratory bird activities and human disturbances have a direct or indirect impact on zooplankton community structure” since, it is not a conclusion derived from the data analysis and results from the current study, rather being a general assumption and thus, I suggest that the authors consider deleting it from the abstract. They can use it to support their results.

Results:
I am not convinced, or at least it is not clearly explained in the manuscript why “The interannual differences of the communities were not found indicating that the seasonal variation in zooplankton community structure in Shahu Lake was much greater than the interannual variation” (lines 265-267). I encourage the authors to provide more details of how they ended up with this very important result.
Also, a table showing the results from one-way ANOVA for biomass and species richness changes among seasons is missing. I believe that the results from lines 202-203: “Zooplankton species richness showed significant seasonal differences (P=0.041)” and line 229 “of zooplankton was significantly lower in winter than in other seasons (P<0.05)” need to be shown in a Table.
The authors have to ensure that the results are presented in a Table and the number of the Table is also included in the text. i.e. line 225: Rotifers density showed no significant seasonal difference (P = 0.123) (Table +++++); The interannual variation in zooplankton density was significant (P = 0.012).
I suggest using either April, July, October, January or spring, summer, autumn, winter, both text, and figures.

Discussion: I believe that the discussion is very weak. The repetition of the results in many parts of the discussion is redundant. There are many linguistic mistakes so in many cases it is unclear what the authors want to point out. The authors are extensively using in their discussion the “Water level fluctuation” as one of the most important factors affecting zooplankton community structure (i.e. Line 381-403) in order to explain the seasonal changes of zooplankton communities in their study. However, they have no data on water level changes in their study in order to use it as an explanatory environmental factor for their findings. This is my major concern, I believe that the authors need to enrich their dataset with more environmental data, and especially water depth, maybe seasonal fluctuations e.c.t. in order to be able to answer their research question 2. I strongly recommend the enrichment of the environmental dataset.

Conclusions
Water environmental factors, water level fluctuations, wintering migratory bird activities, and human disturbances have a direct or indirect impact on zooplankton community structure. This is definitely not a conclusion of the current study since water level fluctuations, wintering migratory bird activities and human disturbances were not tested.

Additional comments

The paper entitled " Seasonal and inter-annual community structure characteristics of zooplankton driven by water environment factors during different hydrological years in a sub-lake of Lake Poyang, China" is interesting and could contribute to a better understanding of the responses of hydrological and water environmental factors to zooplankton community in lakes. I believe, however, that the authors partly answered their second research question since important environmental factors affecting zooplankton community were not included in the study. The authors could also analyzed their dataset by strengthening seasonal and interannual variability examination. Some suggestions would be to estimate species turn-over or estimation of the coefficient of similarity among years/seasons. I strongly recommend a revision by an English native speaker since the linguistic quality of the manuscript is very low and, in many parts, it is not clear what do authors want to say.
Therefore, major revisions are needed so that the manuscript could be considered for publication.

---

## Round 0.2 · Minor Revisions

I can see the authors have addressed the comments well and thus the manuscript is now scientifically acceptable.

However, a final check by Rob Toonen and James Reimer (the Section Editors for this part of the journal) has shown that the English language needs further editing before the manuscript can be finally Accepted. Therefore, we are returning this to you so that you can conduct a final edit of the English language and grammar.

---

## Round 0.3 · accepted · Accept

This is a considerable contribution to the understanding of spatiotemporal zooplankton in Lake Poyang. I can see that the authors have well addressed the reviewers' comments and improved the language and thus it is acceptable for publication.